# Workplace violence and its associated factors among health care workers of a tertiary hospital in Kathmandu, Nepal

**Anupama Bhusal** [1]*, **Apekshya Adhikari** [2], **Pranil Man Singh Pradhan** [3,4]

**1** Central Department of Public Health, Institute of Medicine, Tribhuvan University, Kathmandu, Nepal, **2** School of Health and Society, University of Wollongong, Sydney, Australia, **3** Department of Community Medicine, Maharajgunj Medical Campus, Institute of Medicine, Kathmandu, Nepal, **4** Department of Global Health and Population, Harvard T.H Chan School of Public Health, Boston, MA, United States of America

* bhusalanupama32@gmail.com

**Data Availability Statement:** All relevant data are within the paper and its Supporting Information files.

## Abstract

Workplace violence (WPV) is a globally prevailing public health concerns among healthcare workers. Workplace violence includes occupational abuse (physical, sexual, verbal and psychological), threats or harm among health workers, and workplace harassment. It is important to identify the prevalence of workplace violence at the workplace. Therefore, this study aimed to assess workplace violence and its associated factors among healthcare workers at a tertiary hospital in Kathmandu, Nepal. A descriptive cross-sectional study was carried out among 369 health care workers in a tertiary hospital in Kathmandu. A semi-structured questionnaire was used for data collection. Data was entered and analyzed using SPSS v20. Descriptive statistics were used to assess workplace violence and other independent variables. Bivariate and multivariate logistic regression model was used to examine the factors associated with workplace violence. The prevalence of verbal violence was highest among doctors (34.3%) and nurses (52.8%) followed by bullied/mobbed among doctors (11.9%) and nurses (17%) any time in the past. Experience of any type of workplace violence in the past among doctor was 45.5% and among nurses was 54% while 35.8% doctors and 46.8% nurses had experienced it in the past 12 months. Patients and relatives of patient were major perpetrator for physical and verbal violence while management and staff members were major perpetrators for bullying/mobbing. Participants marital status, work experience, posted department, nature of work shift, frequency of night shift and working hours per week showed statistically significant association with the experience of workplace violence within past 12 months (p<0.05) in binary logistic regression analysis. There is a crucial need to establish evidence-based actions to prevent violence in the workplace and promote a healthy workplace setting. Placing adequate staffs at emergency and medical departments and providing training to cope with the stressful emergency situations would help in minimizing workplace violence among health workers.

**Funding:** The author(s) received no specific funding for this work.

**Competing interests:** The authors have declared that no competing interests exist.

## Introduction

Workplace violence is the act or threat of violence, ranging from verbal abuse to physical assaults directed toward persons at work or on duty. The impact of workplace violence can range from psychological issues to physical injury or even death" [1]. According to the US Occupational Safety and Health Administration (2018), workplace violence is "any act or threat of physical violence, harassment, intimidation or other threatening disruptive behavior that occurs at the work site" [2].

Workplace violence also includes events including occupational abuse, threats or harms among health workers encompassing workplace harassment and abuse that is physical, sexual, verbal, and psychological (WHO) [3]. It is possible to be directly exposed to workplace violence by actual involvement, indirectly exposed through second-hand observation, or both [4]. Workplace elements like insufficient staffing, a severe workload, and inefficient prevention methods raise the risk of exposure to occupational violence and its detrimental effects on patients and health care personnel [5].

Victims who are exposed to traumatic experiences, such as workplace violence, increases their risk of mental health issues [4]. Other effects of workplace violence on its victims are shown by various studies. According to a systematic review of 68 studies involving healthcare employees, being directly exposed to workplace violence is linked to adverse physical, emotional, performance, and economic results in addition to adverse effects on mental health [6].

Violence can happen in any occupational setting and any type of health worker, but healthcare and social support employees are most at risk for non-fatal violence that causes days away from work [1]. Workplace violence has been experienced by 62% of health workers. The most prevalent is verbal abuse (58%), followed by threats (33%) and sexual harassment (12%) [7]. International research shows that between 25% and 67% of nurses experience at least one form of workplace violence every year. A recent study in Canada has shown that nurses are highly exposed to emotional abuse(83%), the threat of assault (78%), followed by physical harm like being hit, being bitten, etc (67%), verbal sexual assault(55%) like unsolicited sexual comments about intimacy and sexual harassment(11%) [5]. In a separate study of 375 healthcare institutions, 30% of medical doctors and 23% of health workers felt insecure in their work environment and reported a high perceived level of threats, attacks, and intimidation [8].

Understanding the different exposure types of occupational violence can help develop prevention strategies and interventions that are better suited to the requirements of health workers [4]. Protecting health workers in the workplace from any form of violence is also a priority of the Government of Nepal for achieving sustainable development goal (SDG) target 8.8 focusing on a healthy workplace [9] and is prioritized by the Health Service Act, 2053 and Safety and Security of health workers and health institutions Act, 2066 under article 114(1) of Constitution of Nepal [10].

This study explores the role that workplace policies, environment and practices have in the occurrence of workplace violence. So the aim of this study is to assess workplace violence and its associated factors among healthcare workers at a tertiary hospital in Kathmandu, Nepal.

## Materials and methods

### Study design and population

This was a hospital-based cross-sectional study conducted from January 5, 2023 to February 3, 2023 in a tertiary hospital in Kathmandu. Bagmati province has 2320 health facilities. There are around 115 hospitals in Kathmandu valley itself which include public, private and academic health institutions, providing both general and specialized healthcare services. There

are 11 central hospitals in the valley. The selected tertiary hospital (Tribhuvan University Teaching Hospital) in this study provides the majority of healthcare services and employs great portion of healthcare workers where 486 doctors and 859 nurses are currently working in this institution. Being a tertiary hospital it receives a high number of patients who are referred from the periphery and require emergency services.

Considering the prevalence of workplace violence as 68% [11]; at a 5% margin of error and 5% level of significance, the final sample size was 369. There was a 100% response rate in this study.

A proportionate random sampling technique was used to select the number of study participants. At first, the tertiary hospital was sampled purposively. The sampling frame of all the healthcare workers employed in the hospital was obtained from hospital administration. Then for the sample selection, groups were formed based on occupation (doctors and nurses) employed in the hospital and a proportionate sample size was calculated for each category to fulfill the required sample size. Sampling was stratified by ward to ensure inclusiveness of healthcare workers working in different departments who interact with different types of patients and provide different services. Study participants were selected from respective hospital wards using a simple random sampling technique.

## Study variables and data collection tools

**Outcome variables.** Workplace violence was the outcome variable of the study. Participants were asked whether they had experienced physical violence, sexual harassment, bullying/mobbing or verbal abuse ever and during the past 12 months. Those who answered "yes" to at least one of these questions were categorized as having experienced workplace violence. Various types of violence were defined in this study as,

- Physical violence: The use of physical force against another person or group that results in physical, sexual or psychological harm. It includes beating, kicking, slapping, stabbing, shooting, pushing, biting, and pinching, among others [12].

- Sexual harassment/violence: Any unwanted, unreciprocated and unwelcome behavior of a sexual nature that is offensive to the person involved, and causes that person to be threatened, humiliated or embarrassed [12].

- Verbal violence: Using inappropriate language, insults and condescending expressions at a workplace [13].

- Bullying/mobbing: Repeated and over time offensive behaviour through vindictive, cruel, or malicious attempts to humiliate or undermine an individual or groups of employees [12].

**Independent variables.** Independent variables were selected through an extensive literature review of similar articles and were broadly classified into socio-demographic variables, work-related variables, and prevention and reporting mechanism.

Socio-demographic variables included age, sex, educational status, marital status and occupation. Work-related factors included work experience, working department, job status, and nature of work shift, number of night shift, sleep pattern and working hours per week [11, 14–17].

**Data collection tool.** The data was collected through a semi-structured questionnaire administered by the researcher. The self-administered questionnaire was divided into two sections:

**Section I**: Socio-demographic and economic characteristics, work-related factors and prevention and reporting mechanism of workplace violence.

Section II: Questions related to workplace violence using a questionnaire developed and validated by the World Health Organization (WHO), International Council of Nurses (ICN), International Labor Office (ILO) and Population Service International (PSI) in 2003 [12].

## Statistical analysis

To ensure the quality of collected data, each questionnaire was reviewed every day after data collection. Collected data were entered and analyzed using Statistical Package for Social Sciences (SPSS v20.0). Caution was maintained during data entry to minimize the errors and thorough cleaning was done.

Descriptive statistics (including means, standard deviations, frequencies and percentage) was calculated for the socio-demographic variables, work- related variables, and workplace violence. The chi-square test was applied to find the association between the dependent variable (workplace violence) and independent variables (age group, sex, educational attainment, marital status, occupation, work experience, posted department, job status, nature of work shift, night shift per month, work hours per week, hours of sleep). P-value less than 0.05 was taken as statistically significant. The adjusted odds ratio was used to measure the strength of the association between variables in multivariate analysis.

In the test of multi-collinearity, none of the variables had tolerance <0.1 and Variance Inflation Factor was found to be <10 which ensured that there is no relationship or interdependence between independent variables themselves. The p-value was less than 0.05 which means that the full model is statistically significant and predicts the dependent variable.

## Ethical consideration

Ethical approval was obtained from the Institutional Review Committee (IRC) of the Institute of Medicine [Ref: 337(6–11)E2 2079/080]. Formal permission was also taken from the hospital administration. Respondents were explained the purpose of the study, written informed consent was taken and respondents were assured that the information would be kept confidential. Personal identity information was not included in the questionnaires and to maintain the anonymity of respondents, unique codes were used during the entire period of study. Respondents were allowed to fully participate on their wishes (voluntary participation) and could withdraw anytime and their choice of not participating in the study was respected. Data collection, handling, and analysis were done solely by the principal investigator, and anonymity and confidentiality were maintained during every step of the research.

## Results

### General and work-related characteristics of study participants

Table 1 describes the distribution of socio-demographic and work-related characteristics by the type of study participants. Out of 369 health care workers in this study, the mean age of respondents was 29.45 years with a standard deviation of 5.23 years. The majority of nurses were below 30 years of age (65.5%) and majority of doctors were greater than 30 years of age (56.7%). In this study, majority of doctors were male (76.1%) while 100% of nurses were female. Majority of doctors were married (57.5%) while more than half nurses were unmarried (50.6%). Almost three-fourth doctors had completed master's degree (73.1%) while majority of nurses had completed bachelor's degree (76.6%).

Most of the doctors and nurses had less than five years of work experience. Among study participants, most of the doctors worked in emergency/medicine/psychiatry unit (44.8%) and most of the nurses were from surgical /burn unit (31.1%). Majority of doctors and nurses were

**Table 1. Distribution of participants by socio-demographic and work-related characteristics (n = 369).**

| Characteristics | Types of providers | |
|---|---|---|
| | Doctor n(%) | Nurse n(%) |
| **Completed age** | | |
| <30 years | 58 (43.3) | 154 (65.5) |
| ≥30 years | 76 (56.7) | 81 (34.5) |
| Mean age ± SD | | 29.45±5.23 |
| **Sex** | | |
| Male | 102 (76.1) | 0 |
| Female | 32 (23.9) | 235 (100) |
| **Marital status** | | |
| Unmarried | 57 (42.5) | 119 (50.6) |
| Married | 77 (57.5) | 116 (49.4) |
| **Educational attainment** | | |
| Diploma | 0 | 48 (20.4) |
| Bachelor | 36 (26.9) | 180 (76.6) |
| Masters and above | 98 (73.1) | 7 (3) |
| **Work experience** | | |
| <5 years | 106 (79.1) | 138 (58.7) |
| ≥ 5 years | 28 (20.9) | 97 (41.3) |
| **Working department** | | |
| Emergency/Medicine/Psychiatry | 60 (44.8) | 59 (25.1) |
| ICU/HDU | 18 (13.4) | 41 (17.4) |
| Obs/Gynae/Maternity/Birthing | 12 (9) | 31 (13.2) |
| Surgical/Burn unit | 33 (24.6) | 73 (31.1) |
| Paediatric unit | 11 (8.2) | 31 (13.2) |
| **Job status** | | |
| Permanent | 17 (12.7) | 56 (23.8) |
| Temporary/contract | 117 (87.3) | 179 (76.2) |
| **Nature of work shift** | | |
| Rotating shift | 91 (67.9) | 222 (94.5) |
| Non-rotating shift | 43 (32.1) | 13 (5.5) |
| **Frequency of night shifts per month** | | |
| < 8 shifts | 34 (25.4) | 75 (31.9) |
| ≥ 8 shifts | 100 (74.6) | 160 (68.1) |
| **Working hours per week** | | |
| ≤ 48 hours | 54 (40.3) | 211 (89.8) |
| >48 hours | 80 (59.7) | 24 (10.2) |
| **Sleep habit/pattern** | | |
| < 7 hours | 76 (56.7) | 39 (16.6) |
| ≥ 7 hours | 58 (43.3) | 196 (83.4) |

on temporary job status and were working on rotating shift. Study showed that doctors (74.6%) worked more than or equal to 8 night shifts per month than nurses (68.1%). Majority of doctors worked more than 48 hours per week and slept less than 7 hours per day than nurses.

## Prevalence of workplace violence

Table 2 shows the prevalence of different types of workplace violence in a healthcare setting by the type of provider. In this study majority of nurses were found to have experienced verbal

**Table 2. Prevalence of different workplace violence among respondents (n = 369).**

| Category | Type of provider | |
|---|---|---|
| | Doctor n(%) | Nurse n(%) |
| **Physical violence** | | |
| Any time in the past | 12 (9) | 36 (15.3) |
| In the past 12 months | 7 (5.2) | 27 (11.5) |
| **Verbal violence** | | |
| Any time in the past | 46 (34.3) | 124 (52.8) |
| In the past 12 months | 38 (28.4) | 106 (45.1) |
| **Bullied/Mobbed** | | |
| Any time in the past | 16 (11.9) | 40 (17) |
| In the past 12 months | 12 (9) | 31 (13.2) |
| **Sexual violence** | | |
| Any time in the past | 5 (3.7) | 16 (6.8) |
| In the past 12 months | 2 (1.5) | 13 (5.5) |
| **Any type of violence experienced** | | |
| Any time in the past | 61 (45.5) | 127 (54) |
| In the past 12 months | 48 (35.8) | 110 (46.8) |

violence any time in the past (52.8%) and 45.1% experienced it in past 12 months. Similarly majority of doctors also experienced verbal violence in the past (34.3%) and in past 12 months (28.4%). Overall, nurses had experienced all types of violence i.e. physical (11.5%), bullied/mobbed (13.2%), and sexual violence (5.5%) more in the past 12 months than doctors where 5.2% experienced physical violence, 9% were bullied/mobbed and 1.5% experienced sexual violence in past 12 months.

## Perpetrators of workplace violence

Table 3 shows the perpetrators of different types of violence that healthcare workers experienced in their workplace. The study showed that majority of doctors experienced physical violence from patients (50%), verbal violence from relatives of patients (50%), bullied/mobbed by management/supervisor (50%) and major perpetrator for sexual violence were relative of

**Table 3. Perpetrators of workplace violence by types of provider (n = 369).**

| Type of provider | Variables | Types of violence [n(%)] | | | |
|---|---|---|---|---|---|
| | | Physical (n = 48) | Verbal (n = 170) | Bully/Mob (n = 56) | Sexual (n = 21) |
| Doctor | Patient/client | 6 (50) | 13 (28.3) | 1 (6.2) | 0 |
| | Relative of patient/client | 5 (41.7) | 23 (50) | 4 (25) | 2 (40) |
| | Staff member | 1 (8.3) | 6 (13) | 1 (6.2) | 1 (20) |
| | Management/supervisor | 0 | 4 (8.7) | 8 (50) | 0 |
| | External colleague/General Public | 0 | 0 | 2 (12.5) | 2 (40) |
| | Total | 12 (100) | 46(100) | 16(100) | 5(100) |
| Nurse | Patient/client | 27 (75) | 47 (37.9) | 7 (17.5) | 1 (6.2) |
| | Relative of patient/client | 6 (16.7) | 52 (41.9) | 13 (32.5) | 5 (31.2) |
| | Staff member | 1 (2.8) | 12 (9.7) | 14 (35) | 6 (37.5) |
| | Management/supervisor | 2 (5.6) | 11 (8.9) | 6 (15) | 2 (12.5) |
| | External colleague/General Public | 0 | 2 (1.6) | 0 | 2 (12.5) |
| | Total | 36(100) | 124(100) | 40(100) | 16(100) |

patients (40%) and external colleague/general public (40%). Similarly for nurses, major perpetrator for physical violence were patients (75%), 41.9% experienced verbal violence from relatives of patients followed by patients themselves (37.9%). Major perpetrator for bullying/mobbing were found to be staff members of same hospital (35%) followed by relatives of patients (32.5%) and also experienced sexual violence from staff members (37.5%) and relatives of patients (31.2%).

## Factors associated with workplace violence

Table 4 shows the factors associated with different types of workplace violence in the past 12 months. The result of the binary logistic regression analysis shows that work experience of less than 5 years (AOR 0.240 95% CI 0.085–0.679), posted in ICU/HDU department (AOR 17.298 95% CI 1.950–153.466) and those working in more than 8 night shifts per month to more likely to experience physical violence.

Similarly, verbal violence was found to be significantly associated with marital status, posted department and nature of work shift. Unmarried respondents (AOR 1.814 95% CI 0.053–3.124), working in emergency/medicine/psychiatry unit (AOR 5.574 95% CI 2.248–13.824), ICU/HDU unit (AOR 3.194 95% CI 1.234–8.266) surgical unit (2.755 95% CI 1.144–6.634) and working in rotating shift (AOR 3.569 95% CI 1.588–8.024) were found associated with verbal violence. Furthermore, being unmarried (AOR 2.576 95% CI 1.097–5.586) was found significantly associated with experience of bullying/mobbing and working for more than 48 hours a week was found significantly associated with the experience of sexual violence.

**Table 4. Factors associated with workplace violence.**

| Variables | Types of violence | | | |
|---|---|---|---|---|
| | Physical AOR (95% CI) | Verbal AOR (95% CI) | Bullied/Mobbed AOR (95% CI) | Sexual AOR (95% CI) |
| **Marital status** | | | | |
| Unmarried | 1.345 (0.512–3.537) | **1.814* (0.053–3.124)** | **2.476* (1.097–5.586)** | 2.038 (0.463–8.970) |
| Married | Ref | Ref | Ref | Ref |
| **Work experience** | | | | |
| < 5 years | **0.240* (0.085–0.679)** | 0.808 (0.420–1.557) | 1.432 (0.529–3.876) | 0.289 (0.061–1.379) |
| ≥ 5 years | Ref | Ref | Ref | Ref |
| **Posted department** | | | | |
| Emergency/ medicine/psychiatry | 7.975 (0.90–70.47) | **5.574* (2.248–13.824)** | 1.90 (0.562–6.424) | 0.817 (0.112–5.942) |
| ICU/HDU | **17.298* (1.950–153.466)** | **3.194* (1.234–8.266)** | 0.540 (0.122–2.385) | 0.486 (0.038–6.214) |
| Obs/Gynae | 1.873 (0.15–23.436) | 1.506 (0.523–4.332) | 2.158 (0.552–8.444) | 1.537 (0.196–12.054) |
| Surgical | 3.933 (0.437–35.37) | **2.755* (1.144–6.634)** | 0.951 (0.269–3.360) | 1.746 (0.299–10.209) |
| Paediatric | Ref | Ref | Ref | Ref |
| **Nature of work shift** | | | | |
| Rotating | 4.740 (0.573–39.18) | **3.569* (1.588–8.024)** | 1.886 (0.585–6.083) | 1.323 (0.129–13.583) |
| Non-rotating | Ref | Ref | Ref | Ref |
| **Frequency of night shift** | | | | |
| <8 shifts | **0.344* (0.119–0.997)** | 0.710 (0.414–1.217) | 0.962 (0.439–2.110) | 0.715 (0.171–2.988) |
| ≥ 8 shifts | Ref | Ref | Ref | Ref |
| **Working hours per week** | | | | |
| ≤ 48 hours | 0.417 (0.140–1.241) | 0.519 (0.258–1.045) | 0.957 (0.348–2.631) | **0.152* (0.038–0.609)** |
| >48 hours | Ref | Ref | Ref | Ref |

Note: *Statistically significant at 5% level of confidence

## Discussion

The prevalence of workplace violence was assessed by adopting and modifying the tool developed and validated by the World Health Organization (WHO), International Council of Nurses (ICN), International Labor Office (ILO), and PSI in 2003 [12]. After analyzing the data received from 369 participants using this tool, the result showed that 50.9% of respondents had experienced any type of workplace violence in the past among which 45.5% doctors and 54% of nurses experienced workplace violence which is similar to the findings of the study in China [18] but is less than that found in a study conducted in Rome among 956 respondents (66.5%) [19]. Similarly, 42.8% had experienced WPV in the past 12 months among which 35.8% doctors and 46.8% nurses experienced workplace violence which is a similar finding to the study conducted in Malaysia [20]. This finding is more than that conducted in China among GPs [21] and less than the prevalence found in a study conducted among nurses in Nepal [11], Kenya [17], and a systematic analysis conducted in India [22]. The reason behind such variation can be difference of research subjects. Some of the study has taken general practitioners only [21] as their research subject while many others have taken nurses only [4, 23–27]. This study included both doctors/residents and nurses who work more closely with patients for longer hours than studies including only general practitioners who have less close contact with patients or only nurses in which prevalence would be much higher.

Among various forms of violence, verbal violence was experienced more by doctors (34.3%) and nurses (52.8%) which is lower than the findings of the study conducted in Italy [28], Kenya [17], Tunisia [29], and Cyprus Republic [30]. Similar finding on verbal violence was observed in the study conducted in Italy [31]. This difference in the experiences of different forms of violence might be differences in the health care system and study setting, personal characteristics of the study population, and cultural differences.

Perpetrators of the workplace violence were mainly relatives of patient/clients for verbal violence and bullying/mobbing for both doctors and nurses. Sexual violence was found perpetrated by relative of patient and external colleague among doctors and staff members of same hospital were found major perpetrator of sexual violence among nurses. Major physical abuses were found to be perpetrated by clients themselves as patients and their relatives frequently contact health workers to receive treatment and follow up. Longer waiting hours, quality of service provided, etc are found to be the reason behind perpetrating violence in the healthcare setting. Similar findings were seen in studies conducted in China [15] and Nepal [11].

In this study, there is no significant association between the age, sex of the respondent, educational attainment, occupation, and hours of sleep with WPV. This finding is similar to the study done in Nepal [14]. A study done in Malaysia has shown that nurses aged 30 years and younger are more likely to experience WPV than those who are greater than 30 years [20] and a systematic review and meta-analysis done in India showed that younger health professionals are more at risk of experiencing WPV than older/senior colleagues [22]. This finding is in contrast to the study conducted in Nepal [11], Norway [32], China [21], and the USA [33]. Nurses are found to experience more workplace violence than doctors in studies conducted in China [15] and Italy [28, 31] which is similar to the finding of this study. Nurses are in the frontline position to directly interact with patients and their relatives and are at higher risk of violence. In contrast to this doctors were found more likely to experience violence in a study conducted in Bangladesh [34]. Sex was found significant in the study conducted in the USA [35] and reports from China shows educational attainment to be significantly associated with WPV where general practitioners with higher level of education were more likely to experience non-physical violence [21, 36]. Job type, employment pattern and job experience were not found significant in the study conducted in primary hospitals in China [37] which is similar to the findings of this study.

In a binary logistic regression conducted in this study, marital status, work experience, posted department, nature of work shift, frequency of night shift and working hours per week were found to be associated with different types of workplace violence within the past 12 months. In this study, unmarried/single respondents were 1.814 times more likely to experience verbal violence (95% CI 0.053–3.124) and 2.476 times more likely to be bullied/mobbed (95% CI 1.097–5.586) which coincides with the study in Malaysia [20] where single participants are more likely to experience WPV while married participants are found more at risk of experiencing WPV in a study conducted in Nepal [11] and China [37]. In contrast to this finding, marital status is not found associated with WPV in a study done among nurses in Iran [26]. Unmarried respondents being vulnerable to experience violence may be due to less work experience, inexperience in dealing with stressful situations, long hours of work and shift work. Similarly posted department was found significantly associated with WPV in this study where health workers posted in the emergency/medicine/psychiatry department/unit were 5.574 times more likely to experience verbal violence (95% CI 2.248–13.824) which is similar to the finding of study in Malaysia [20], USA [38] and systematic and meta-analysis where six studies reported emergency departments as a most common place for WPV [22]. A study done in Norway [32] has shown the highest prevalence of WPV in the psychiatric department. Similarly physical and verbal violence was found more likely to be experienced by those working in ICU/HDU and surgical units. The reason behind the higher odds of experiencing WPV in the emergency/medicine/psychiatry unit, ICU/HDU and surgical unit may be due to high patient flow, the seriousness of patient's condition, high workload, longer hours of contact with patient and their relatives, and high stress of health workers. In a study conducted in Bangladesh among health care workers, those involved in shift work were found to be at 1.52 times higher risk of experiencing workplace violence [34] which is a similar finding as this study where participants working in rotating shift were 3.569 times more likely to experience verbal violence (95% CI 1.588–8.024) than those who worked in non-rotating shift. A study in Macau, China has shown that working in shift work and night duty are highly associated with physical abuse [15]. A study conducted in Iran has shown that most of the physical (57%) and verbal violence (35.2%) to occur in the night shifts [26] which is similar to the findings of this study where physical violence was found more associated with higher frequency of night shifts which can be brought on by an increase in patient referrals to emergency rooms and the lack of a management team. Working more hours per week is found to be associated with sexual violence which may be related to the more time spent in workplace, shame and embarrassment and lack of consequences following a sexual harassment report [39].

Therefore following are the suggestions for minimizing the widespread problem of workplace violence, First, formulating and reviewing existing standards and guidelines on any form of misconduct in the workplace by and to healthcare workers and orientation to healthcare workers on the existing complaint reporting mechanism of the institution and encourage them to report a complaint if they ever experienced any form of violence. Secondly, take appropriate action on the complaint regarding experiences of violence especially on sexual violence which will motivate other victims to register the complaint and the perpetrator/abuser to refrain from such activities. Thirdly, identifying and implementing the various coping strategies, offering counselling or other support to the victims and raising mental health awareness among health professionals. Health institutions must ensure that work-load and working hours are within the capacity of health workers which helps in clear decision making. Since unmarried respondents, those working in emergency/medicine/psychiatry and ICU/HDU departments, working in rotating shifts and night shifts are found to experience more violence; the recommended activities should be carried out with special focus to such health workers. Placing adequate staffs at emergency and medical departments and providing training to cope

with the stressful emergency situations would also help in minimizing workplace violence among health workers.

There are some limitations to this study. Since this study was carried out in a single tertiary hospital located in an urban area, the extrapolation of conclusions at the national level could be challenged. Selection of more hospitals as study site would provide better results as experience of violence would vary by work atmosphere, type and size of patients and staffs working at different healthcare setting. Furthermore, doctors and nurses were only taken as study population so the representativeness of findings to all health care workers is limited. Information regarding frequency, and intensity of violence and how it affects functions, health and social well-being of the victims and indirectly affects their patient service were also not included in the study. Experience of workplace violence was assessed retrospectively and there was a chance of recall bias however it was minimized by assessing the experience of workplace violence within the past 12 months which also provided information on recency of experience of violence. Exposure to violence being a sensitive topic there was a chance of respondent bias which was minimized by maintaining the privacy and confidentiality of participants during data collection, analysis, and result dissemination. It was a cross-sectional study revealing the status of the research variables at a certain time, thus the interpretation of the causal relationship between the variables was limited. It will be of benefit to confirm the causality with longitudinal data in future studies.

## Conclusion

This study has shown higher prevalence of workplace violence which is one of the major public health concerns among healthcare workers. This study showed that job status, nature of work-shift, workload, long working hours and frequent exposure to clients contributes to workplace violence. Similarly, marital status, posted departments, work experience, nature of work shift, frequency of night shift and working hours per week were found significantly associated with the experience of workplace violence by healthcare workers. This showed a crucial need to establish evidence-based actions to prevent violence in the workplace, and promote a healthy workplace setting and overall health of the healthcare workers. Initiatives to protect health workers in their own institution with special measures for unmarried respondents, working in emergency/medicine departments and those working in shift work should be focused. Thus, understanding workplace violence and related factors like the posted department and nature of work-shift among healthcare workers can aid the decision-makers at the hospital level and policy makers in Nepalese health system to address workplace problems; design and implement effective and resilient strategies.

## Supporting information

**S1 Checklist. STROBE statement—checklist of items that should be included in reports of observational studies.**
(DOCX)

**S1 Text. Violence dataset.**
(SAV)

## Acknowledgments

We would like to express our deep gratitude to Prof. Dr. Amod Kumar Poudyal and Mr Susan Man Shrestha for their valuable support and suggestions. We would also like to acknowledge the support from the hospital administration for providing us permission for data collection,

and different departments and wards for coordinating with staff members aiding data collection at the study site.

## Author Contributions

**Conceptualization:** Anupama Bhusal, Apekshya Adhikari, Pranil Man Singh Pradhan.

**Data curation:** Anupama Bhusal.

**Formal analysis:** Anupama Bhusal.

**Methodology:** Anupama Bhusal, Pranil Man Singh Pradhan.

**Resources:** Anupama Bhusal.

**Supervision:** Anupama Bhusal, Pranil Man Singh Pradhan.

**Validation:** Anupama Bhusal, Pranil Man Singh Pradhan.

**Writing – original draft:** Anupama Bhusal.

**Writing – review & editing:** Apekshya Adhikari, Pranil Man Singh Pradhan.

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
