## [Decision Letter · Decision Letter 0]

17 May 2023

PONE-D-23-09721Workplace violence and its associated factors among health care workers of a tertiary hospital in Kathmandu, NepalPLOS ONE

Dear Dr. Bhusal,

Thank you for submitting your manuscript to PLOS ONE. After careful consideration, we feel that it has merit but does not fully meet PLOS ONE’s publication criteria as it currently stands. Therefore, we invite you to submit a revised version of the manuscript that addresses the points raised during the review process.I would suggest authors to make necessary changes as per the suggestions from reviewers.  Please submit your revised manuscript by Jul 01 2023 11:59PM. If you will need more time than this to complete your revisions, please reply to this message or contact the journal office at plosone@plos.org. Please include the following items when submitting your revised manuscript:A rebuttal letter that responds to each point raised by the academic editor and reviewer(s). You should upload this letter as a separate file labeled 'Response to Reviewers'.A marked-up copy of your manuscript that highlights changes made to the original version. You should upload this as a separate file labeled 'Revised Manuscript with Track Changes'.An unmarked version of your revised paper without tracked changes. You should upload this as a separate file labeled 'Manuscript'.

We look forward to receiving your revised manuscript.

Kind regards,

Kshitij Karki, MPH, MA

Academic Editor

PLOS ONE

Journal Requirements:

Additional Editor Comments:

Please address the comments of the reviewers.

Reviewers' comments:

Reviewer's Responses to Questions

**Comments to the Author**

1. Is the manuscript technically sound, and do the data support the conclusions?

Reviewer #1: Yes

Reviewer #2: Yes

2. Has the statistical analysis been performed appropriately and rigorously? 

Reviewer #1: No

Reviewer #2: Yes

3. Have the authors made all data underlying the findings in their manuscript fully available?

Reviewer #1: Yes

Reviewer #2: Yes

4. Is the manuscript presented in an intelligible fashion and written in standard English?

Reviewer #1: Yes

Reviewer #2: Yes

5. Review Comments to the Author

Reviewer #1: - The aims of the study must be presented in a clearer way.

- in the paragraph describing the Outcome variables, verbal violence is not clearly reported

- as far as concerns the selection of independent variables, the authors stated they "were selected through an extensive literature review of similar articles". Give relevant bibliometric references.

- In Table 5 the Authors presented the Factors associated with workplace violence, using all the type of violence toghether. I suggest to add also the analyses for the single type of violence, excluding the non-adjusted analysis.

- concerning the comparison with other countries, the authors mentioned Italy, including data of Ielapi et al, involving 203 HCWs. I suggest to use also these references 10.23749/mdl.v110i2.7807; 10.3390/ijerph19116938; 10.1016/j.puhe.2022.04.008

Reviewer #2: This is an interesting paper, looking at the prevalence and factors associated with workplace violence among doctors and nurses in a tertiary hospital in Nepal. The manuscript is well written. However, there are several limitations in terms of methodology and results, making the paper limited of use for public health.

1. Study setting: As this study selected Tribhuvan University Teaching Hospital, which is a large tertiary hospital as the study setting, the representativeness of study sample to all healthcare workers is thus limited. Work atmosphere, types and size of patients and services delivered and staff are different by level of healthcare setting. This can also be associated with workplace violence and confound with the relationship between exposure and outcome. Actually, it would be better if other levels of hospitals were selected as well. The authors should discuss this point as a limitation, including how it affects the study findings.

2. Sampling method: Please describe if the sampling was stratified by ward or hospital department or not. Different ward settings make different types of patients, relatives and services. This can be differently related to violence as well.

3. Outcome variable: I wonder if the frequency, intensity and recency of violence and how it affects functions, health and social well-being of the victims and indirectly affects their patient service were asked. It would be more useful for prevention of workplace violence and provision of help to those victims if such information was included in the questionnaire.

4. Exposure variables: It is well documented that alcohol and substance use (e.g., cannabis, methamphetamine) and psychiatric disorders are related with violence. The most common sources of violence in this study were from patients and their relatives. It would be more useful if we have data on their alcohol and substance use and psychiatric problems. Did the authors collect data relating to type of patients involved in the violence act? I don't see in the table if psychiatric ward and outpatient department were included in the sampling frame. Is there no such department in Tribhunyan Hospital?

5. Results: Tables 1 and 2 could be combined, of which data could be displayed by type of workers. Tables 3 and 4 should also be displayed by type of workers. It would be useful to see the cross-tabulation between types and sources of violence as interventions given to victims would be different by types and sources of violence. (For example, if violence is caused by patients, the intervention would be different to that caused by his/her colleagues or supervisor.) The results of factors associated with violence are not much of use for prevention and supportive interventions to the victims. Apart from posted department, other factors are hardly modified. If there were more details on the characteristics of violence and sources and the impacts on the affected healthcare workers, more specific interventions could be offered.

6. Discussion. The Recommendations suggested by the authors were too general. It is not needed to do this study to provide such recommendations.

6. PLOS authors have the option to publish the peer review history of their article (what does this mean?). If published, this will include your full peer review and any attached files.

Reviewer #1: No

Reviewer #2: No

---

## [Author Response · Author response to Decision Letter 0]

27 Jun 2023

Reviewer 1: 

Comment 1: The aims of the study must be presented in a clearer way.

Response: The aim of this study is to assess workplace violence and its associated factors among healthcare workers at a tertiary hospital in Kathmandu, Nepal. Page number 4, line number 103-105.

Comment 2: In the paragraph describing the Outcome variables, verbal violence is not clearly reported.

Response: Verbal violence: Using inappropriate language, insults and condescending expressions at a workplace. Page number 5, line number 142-143.

Comment 3: As far as concerns the selection of independent variables, the authors stated they "were selected through an extensive literature review of similar articles". Give relevant bibliometric references.

Response: References are added to the manuscript in page number 5, line number 153.

Comment 4: In Table 5 the Authors presented the Factors associated with workplace violence, using all the type of violence together. I suggest to add also the analyses for the single type of violence, excluding the non-adjusted analysis.

Response: Adjusted analysis for single types of violence has been added in the manuscript in page number 11-12.

Comment 5: Concerning the comparison with other countries, the authors mentioned Italy, including data of Ielapi et al, involving 203 HCWs. I suggest to use also these references 10.23749/mdl.v110i2.7807; 10.3390/ijerph19116938; 10.1016/j.puhe.2022.04.008

Response: Information and references added to the manuscript in page number 12, line number 266-267; page number 13, line number 280-281, page 13, and line number 302.

Reviewer 2:

Comment 1: Study setting: As this study selected Tribhuvan University Teaching Hospital, which is a large tertiary hospital as the study setting, the representativeness of study sample to all healthcare workers is thus limited. Work atmosphere, types and size of patients and services delivered and staff are different by level of healthcare setting. This can also be associated with workplace violence and confound with the relationship between exposure and outcome. Actually, it would be better if other levels of hospitals were selected as well. The authors should discuss this point as a limitation, including how it affects the study findings.

Response: Thank you for the suggestion. We have now added this point in the limitation “Selection of more hospitals as study site would provide better results as experience of violence would vary by work atmosphere, type and size of patients and staffs working at different healthcare setting. Furthermore, doctors and nurses were only taken as study population so the representativeness of findings to all health care workers is limited” in page number 15, line number 368-371.

Comment 2: Sampling method: Please describe if the sampling was stratified by ward or hospital department or not. Different ward settings make different types of patients, relatives and services. This can be differently related to violence as well.

Response: Sampling was stratified by ward to ensure inclusiveness of healthcare workers working in different departments who interact with different types of patients and provide different services. This has been added in the manuscript in page 4, line number 126-128.

Comment 3: Outcome variable: I wonder if the frequency, intensity and recency of violence and how it affects functions, health and social well-being of the victims and indirectly affects their patient service were asked. It would be more useful for prevention of workplace violence and provision of help to those victims if such information was included in the questionnaire.

Response: Information regarding frequency, and intensity of violence and how it affects functions, health and social well-being of the victims and indirectly affects their patient service were not included in the study. However the information regarding recency was received by asking question “Experienced workplace violence in the past 12 months”. This information has been added in the manuscript in page number 15, line number 374- 377.

Comment 4: Exposure variables: It is well documented that alcohol and substance use (e.g., cannabis, methamphetamine) and psychiatric disorders are related with violence. The most common sources of violence in this study were from patients and their relatives. It would be more useful if we have data on their alcohol and substance use and psychiatric problems. Did the authors collect data relating to type of patients involved in the violence act? I don't see in the table if psychiatric ward and outpatient department were included in the sampling frame. Is there no such department in Tribhuvan Hospital?

Response: Information regarding the use of alcohol, substance use and psychiatric disorder of patients was not collected in this study as this study provides information taken only from health care workers. Since the sample were taken in hospital wards, type of patient were also not specified in the study. Data was taken from health care workers working in psychiatric ward but due to the less number of healthcare workers in the psychiatric ward it was categorized with other wards having similar nature of work for further analysis. It is shown in table 1 and 4.

Comment 5: Results: Tables 1 and 2 could be combined, of which data could be displayed by type of workers. Tables 3 and 4 should also be displayed by type of workers. It would be useful to see the cross-tabulation between types and sources of violence as interventions given to victims would be different by types and sources of violence. (For example, if violence is caused by patients, the intervention would be different to that caused by his/her colleagues or supervisor.) The results of factors associated with violence are not much of use for prevention and supportive interventions to the victims. Apart from posted department, other factors are hardly modified. If there were more details on the characteristics of violence and sources and the impacts on the affected healthcare workers, more specific interventions could be offered.

Response: Table 1 and 2 have been combined and all the tables are displayed by the type of workers. Factors associated with violence has also been shown by different types of violence. These changes has been made in table 1, 2, 3 and 4.

Comment 6: Discussion. The Recommendations suggested by the authors were too general. It is not needed to do this study to provide such recommendations.

Response: “Initiatives to protect health workers in their own institution with special measures for unmarried respondents, working in emergency/medicine departments and those working in shift work should be focused. Thus, understanding workplace violence and related factors like the posted department and nature of work-shift among healthcare workers can aid the decision-makers at the hospital level and policy makers in Nepalese health system to address workplace problems; design and implement effective and resilient strategies.”

These recommendations are made according to the findings of the study as marital status, posted department and nature of work shift are found significantly associated with workplace violence in page number 16, line number 392-398.

---

## [Decision Letter · Decision Letter 1]

2 Jul 2023

Workplace violence and its associated factors among health care workers of a tertiary hospital in Kathmandu, Nepal

PONE-D-23-09721R1

Dear Dr. Anupama Bhusal,

We’re pleased to inform you that your manuscript has been judged scientifically suitable for publication and will be formally accepted for publication once it meets all outstanding technical requirements.

Kind regards,

Kshitij Karki, MPH, MA

Academic Editor

PLOS ONE

Additional Editor Comments (optional):

Reviewers' comments:

Reviewer's Responses to Questions

**Comments to the Author**

1. If the authors have adequately addressed your comments raised in a previous round of review and you feel that this manuscript is now acceptable for publication, you may indicate that here to bypass the “Comments to the Author” section, enter your conflict of interest statement in the “Confidential to Editor” section, and submit your "Accept" recommendation.

Reviewer #1: All comments have been addressed

2. Is the manuscript technically sound, and do the data support the conclusions?

Reviewer #1: Yes

3. Has the statistical analysis been performed appropriately and rigorously? 

Reviewer #1: Yes

4. Have the authors made all data underlying the findings in their manuscript fully available?

Reviewer #1: Yes

5. Is the manuscript presented in an intelligible fashion and written in standard English?

Reviewer #1: Yes

6. Review Comments to the Author

Reviewer #1: The authors made the requested changes according to the reviewer's suggestion. According to me the manuscript is suitable for publication.

7. PLOS authors have the option to publish the peer review history of their article (what does this mean?). If published, this will include your full peer review and any attached files.

Reviewer #1: No

---

## [Editor Report · Acceptance letter]

19 Jul 2023

PONE-D-23-09721R1 

Workplace violence and its associated factors among health care workers of a tertiary hospital in Kathmandu, Nepal 

Dear Dr. Bhusal:

I'm pleased to inform you that your manuscript has been deemed suitable for publication in PLOS ONE. Congratulations! Your manuscript is now with our production department. 

Kind regards, 

on behalf of

Dr. Kshitij Karki 

Academic Editor

PLOS ONE